# De novo design of a homo-trimeric amantadine-binding protein

Jooyoung Park[1,2†*], Brinda Selvaraj[3], Andrew C McShan[4], Scott E Boyken[1,2‡], Kathy Y Wei[1,2,5], Gustav Oberdorfer[6], William DeGrado[7], Nikolaos G Sgourakis[4], Matthew J Cuneo[3,8], Dean AA Myles[3], David Baker[1,2*]

[1]Department of Biochemistry, University of Washington, Seattle, United States; [2]Institute for Protein Design, University of Washington, Seattle, United States; [3]Neutron Sciences Directorate, Oak Ridge National Laboratory, Oak Ridge, United States; [4]Department of Chemistry and Biochemistry, University of California, Santa Cruz, Santa Cruz, United States; [5]Department of Bioengineering, University of California, Berkeley, Berkeley, United States; [6]Institute of Biochemistry, Graz University of Technology, Graz, Austria; [7]Department of Pharmaceutical Chemistry, University of California, San Francisco, San Francisco, United States; [8]Department of Structural Biology, St. Jude Children's Research Hospital, Memphis, United States

**Abstract** The computational design of a symmetric protein homo-oligomer that binds a symmetry-matched small molecule larger than a metal ion has not yet been achieved. We used de novo protein design to create a homo-trimeric protein that binds the $C_3$ symmetric small molecule drug amantadine with each protein monomer making identical interactions with each face of the small molecule. Solution NMR data show that the protein has regular three-fold symmetry and undergoes localized structural changes upon ligand binding. A high-resolution X-ray structure reveals a close overall match to the design model with the exception of water molecules in the amantadine binding site not included in the Rosetta design calculations, and a neutron structure provides experimental validation of the computationally designed hydrogen-bond networks. Exploration of approaches to generate a small molecule inducible homo-trimerization system based on the design highlight challenges that must be overcome to computationally design such systems.

*For correspondence:
e.jooyoungpark@gmail.com (JP);
dabaker@uw.edu (DB)

Present address: †Sana Biotechnology, Inc, Seattle, United States; ‡Lyell Immunopharma, Inc, Seattle, United States

## Introduction

While there has been progress in the de novo design of small molecule binding proteins (*Tinberg et al., 2013*; *Ollikainen et al., 2015*; *Mills et al., 2016*; *Polizzi et al., 2017*; *Dou et al., 2018*), there are still considerable challenges in this area (*Dou et al., 2017*). There has also been progress in designing protein structures with internal symmetry (*Boyken et al., 2016*; *Ghirlanda et al., 2004*). We focus in this paper on the challenge of designing symmetric protein homo-oligomers that bind to symmetry matched small molecules such that each protein monomer makes identical interactions with the small molecule. From the protein design standpoint, this challenge is interesting as it enables more economical design strategies in which one protein-small molecule interface is utilized multiple times, analogous to the use of a single designed protein-protein interface in self-assembling protein nanostructures (*Bale et al., 2016*; *Hsia et al., 2016*). From the applications standpoint, the challenge has considerable importance because it provides a stepping stone to ligand induced homo-oligomerization systems, which are increasingly in demand in cellular engineering applications (*Fegan et al., 2010*; *DeRose et al., 2013*). Chemically-inducible dimerization systems (*Spencer et al., 1996*) have been utilized to modulate signal transduction (*Spencer et al., 1996*; *Mallet et al., 2002*; *Guerrero et al., 2008*), transcriptional activation

(*Nyanguile et al., 1997*), and post-translational modification (*Stankunas et al., 2003*), and as components for logic gates (*Miyamoto et al., 2012*). However, to our knowledge, no chemically-inducible trimerization systems have been developed despite the importance of trimerization in pro-apoptotic and pro-inflammatory signaling cascades (*Spencer et al., 1996*).

## Results

### Computational design strategy for ABP

We set out to design trimeric proteins that bind small molecules with three-fold symmetry on their symmetry axes. We focused on the $C_3$ symmetric compound amantadine as it is an FDA approved drug (https://www.accessdata.fda.gov/drugsatfda_docs/label/2009/016023s041,018101s016lbl.pdf) with a low side effect profile (*Perez-Lloret and Rascol, 2018*). To de novo design amantadine-binding sites at the protein trimer $C_3$ axes, we started from parametrically generated $C_3$ symmetric helical bundle backbones consisting of two concentric rings each with three helices. The symmetry axes of the protein scaffold and the amantadine were aligned, and the remaining two degrees of freedom (the placement along the symmetry axis, and the rotation around this axis) were sampled by grid search (*Figure 1a*). For each placement, RosettaDesign was used to optimize the identities and conformations of the residues within 12.5 Å of the amantadine for high affinity binding, and residue conformations at distances farther than 12.5 Å to retain hydrogen-bond networks identified by Rosetta HBNet (*Figure 1b–c*). The search was restricted to symmetric solutions in which each monomer is identical in sequence and structure. We found a particularly low energy solution starting from a previously characterized design with a high-resolution crystal structure (2L6HC3_13) (*Boyken et al., 2016*) (*Figure 1a and d*). This solution, which we refer to as ABP (**a**mantadine-**b**inding **p**rotein), contains 19 amino acid changes compared to 2L6HC3_13 (*Figure 1d*). The interactions critical for amantadine binding include hydrogen bonds from Ser-71 to the polar amino group of amantadine and a shape complementary binding pocket composed by Ile-64, Leu-67, and Ala-68 (*Figure 1b*).

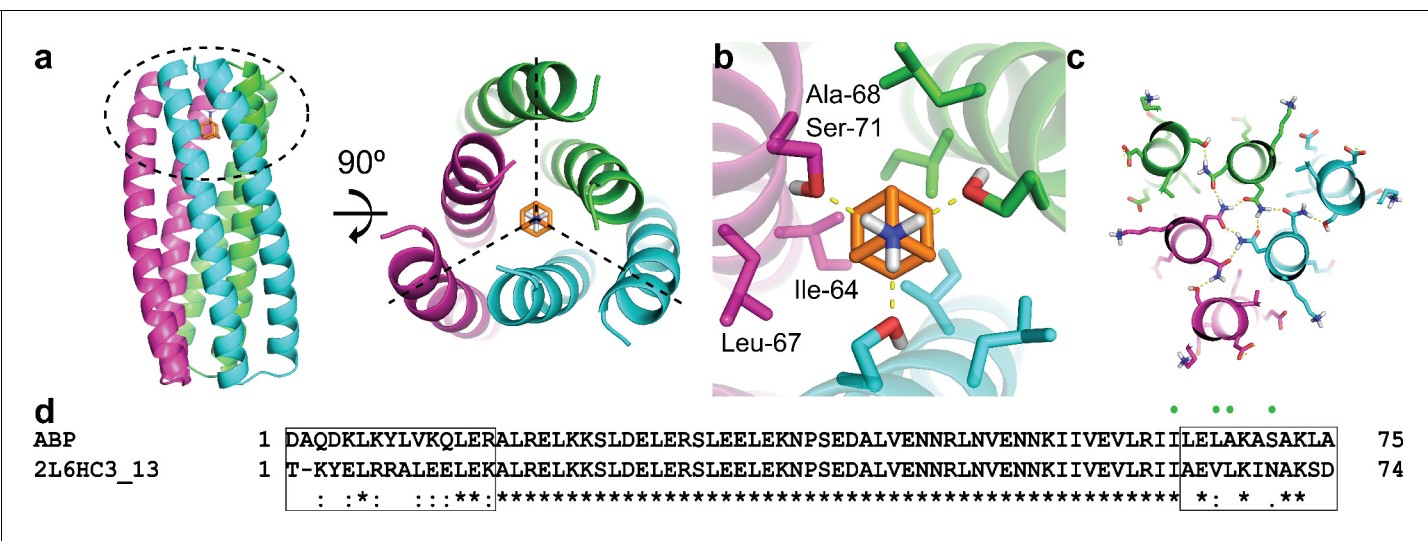

**Figure 1.** Computational design methodology. (**a**) The homo-trimeric scaffold was designed to bind amantadine such that the $C_3$ axes of the protein and the small molecule are aligned. Amantadine is colored orange and each monomer of ABP is colored magenta, green, or cyan. The amantadine binding site is highlighted by a dashed oval. (**b**) The binding pocket in ABP was designed to have polar serine residues (Ser-71) that hydrogen-bond (yellow dashed lines) to the amino group of amantadine and nonpolar residues (Ile-64, Leu-67, and Ala-68) to complement the shape of the hydrophobic moiety of amantadine. (**c**) The design model contains hydrogen-bond networks that specify the trimeric assembly of ABP. (**d**) A sequence alignment of ABP and 2L6HC3_13 is shown, with mutated regions shown in black boxes. Both sequences are preceded by a five-residue GHSMG pre-sequence (not shown) that result from the cloning strategy. The residues highlighted in (**b**) are annotated by green circles.

The online version of this article includes the following figure supplement(s) for figure 1:

**Figure supplement 1.** ABP variant designs.

## Binding interaction of amantadine with ABP

A synthetic gene encoding ABP was obtained and the protein expressed in *E. coli*. The design was expressed at high levels in the soluble fraction and was found by SEC-MALS to be a trimer in the presence and absence of amantadine (*Figure 2a*). Interactions with amantadine were probed using thermofluor dye binding assay (differential scanning fluorimetry). The thermofluor melting curve for apo-ABP exhibited a high initial fluorescence signal at 25° C (*Figure 2b*), indicating that hydrophobic residues in the protein core are exposed to solvent, characteristic of a molten globule state. As the protein was heated to 95° C, the fluorescence signal decreased, likely due to aggregation and/or complete unfolding. In the presence of amantadine (1 mM), the initial fluorescence signal was much lower, characteristic of properly folded proteins (*Figure 2b*), suggesting that amantadine binding may cause local ordering and exclude solvent. In contrast, 2L6HC3_13, which has the same backbone parameters but lacks the amantadine binding site, is thermally stable by thermofluor assay, only starting to denature at ~80° C (*Figure 2b*), consistent with previous work (*Boyken et al., 2016*). As expected, amantadine had no effect on the melting curve of 2L6HC3_13, suggesting the interactions with ABP are through the designed binding site (*Figure 2b*). The circular dichroism (CD)

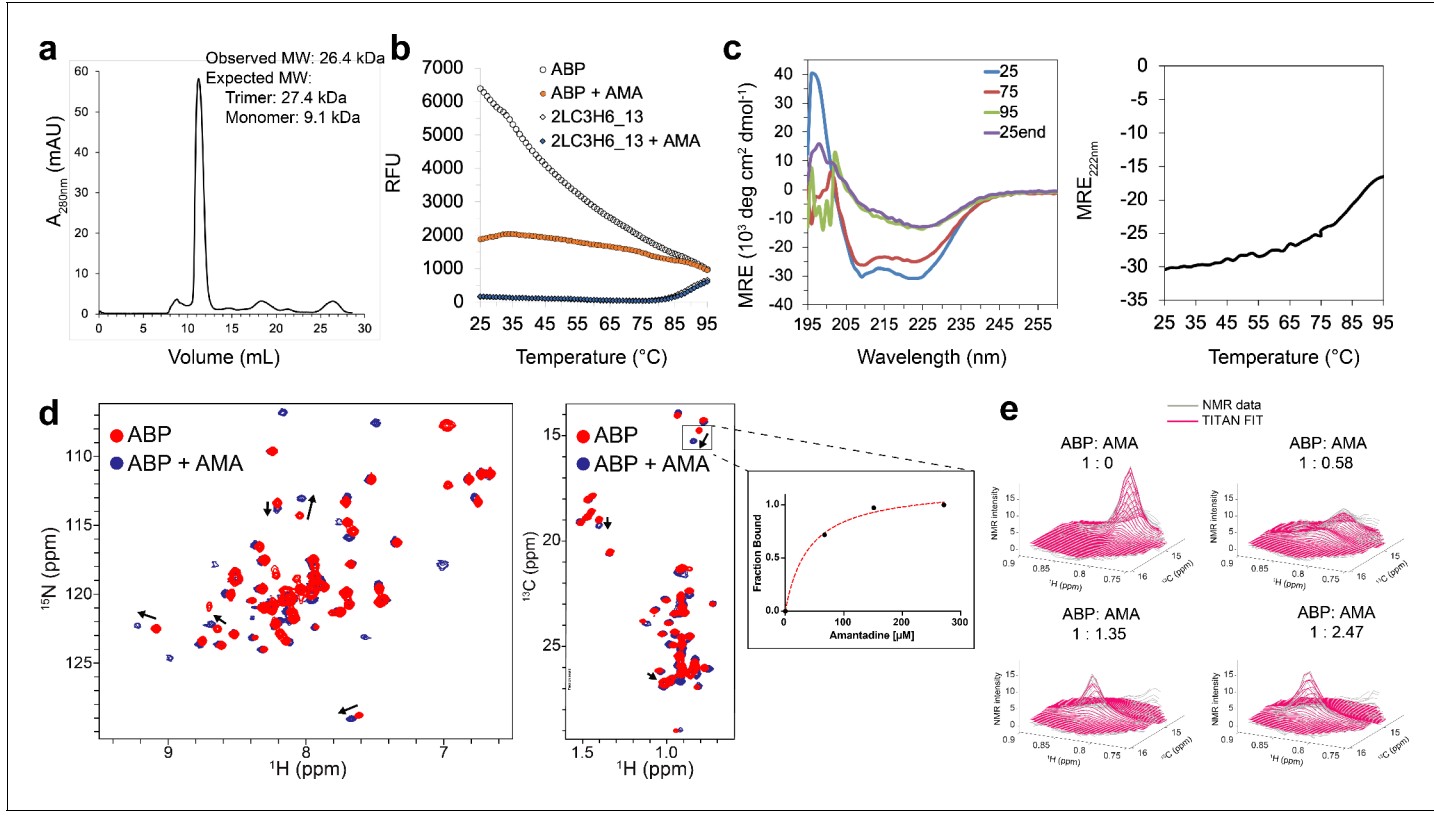

**Figure 2.** Binding characterization of amantadine to ABP. (**a**) SEC chromatogram monitoring absorbance at 280 nm (mAU) and estimated molecular mass (from MALS). (**b**) Apo-ABP (orange, open circle) exhibits a high initial fluorescence signal that is lowered in the presence of amantadine (orange, solid circle). As expected, 2LC3H6_13 (blue, open triangle) and 2LC3H6_13 plus amantadine (blue, solid triangle) exhibit a very low initial fluorescence signal and overlap almost identically. (**c**) The CD spectrum of ABP at 25°C, 75°C, 95°C, and 25°C after heating and cooling. The CD spectrum of ABP at 25°C suggests an all α-helical structure that remains fairly stable up to 75°C. (**d**) 2D amide $^1$H-$^{15}$N HMQC spectra (left) and 2D methyl $^1$H-$^{13}$C HMQC spectra (right) of 250 μM apo-ABP (red) or ABP in the presence of 2 mM amantadine (blue) recorded at 800 MHz, 37°C. Titration of amantadine leads to significant changes in the ABP NMR spectra (arrows). To the right of the 2D methyl methyl $^1$H-$^{13}$C HMQC spectra an inset of dissociation constant estimate through conventional fraction bound analysis is shown for the affected ILE methyl group, with an estimated $K_D$ of <55 μM. (**e**) NMR line shape fitting of ABP throughout the NMR titration with amantadine performed in the program TITAN using a two-state binding model for the affected ILE methyl group. The NMR data (gray) are shown versus the TITAN fit (magenta).

The online version of this article includes the following figure supplement(s) for figure 2:

**Figure supplement 1.** CD spectrum of ABP in the presence amantadine.

**Figure supplement 2.** NMR titration of ABP with amantadine.

spectrum of ABP at 25° C suggests an all α-helical structure, with negative bands at 222 nm and 208 nm, and a positive band at 190 nm (*Figure 2c*). As the sample was heated to 95° C, a loss in CD signal was observed which was not significantly altered in the presence of 1 mM amantadine (*Figure 2c* and *Figure 2—figure supplement 1*).

## Solution NMR analysis of amantadine binding

To examine whether the hydrophobic residues (Ile-64, Leu-67, and Ala-68) contacting amantadine (*Figure 1b*) undergo conformational changes upon ligand binding, as suggested by the thermofluor assays, ABP was selectively $^{13}$C-methyl-labeled at Ala, Ile, Leu, and Val residues in a $^{12}$C/$^{15}$N/$^{1}$H background and characterized by solution NMR, both in the presence and absence of amantadine. We observe resonances for 70 out of 79 amide and 51 out of 51 methyl groups present in the primary sequence; a small number of amide NMR resonances are missing likely due to conformational exchange-induced line broadening. The single set of peaks for all three polypeptide chains of the apo-ABP suggests that it populates a homogeneous and symmetric ensemble of conformers in solution (*Figure 2d*). The $^{1}$H dispersion in the 2D $^{1}$H-$^{15}$N HQMC and 2D $^{1}$H-$^{13}$C HMQC NMR spectra suggests that ABP, both in the presence and absence of amantadine, adopts a similar helical structure (*Figure 2d*). A full titration of amantadine on ABP confirms the formation of a stable amantadine-ABP complex, with chemical exchange between the free and bound ABP states slow on the NMR chemical shift time scale (residence time of 10–100 milliseconds) (*Figure 2d*, arrows). An NMR line shape fitting of the three most significantly affected ABP methyl resonances (>0.1 ppm chemical shift deviation between free and bound states) using TITAN suggest an apparent dissociation constant ($K_D$) of 24.1 ± 2.7 μM and upper limit for off-rate constant ($k_{off}$) of 60.7 ± 5.6 s$^{-1}$ (on-rate constant of 2.5 × 10$^{6}$ M$^{-1}$ sec$^{-1}$) (*Figure 2e* and *Figure 2—figure supplement 2*). The relatively slow fitted on- and off-rate constants are consistent with a buried amantadine binding site. Fixing the $K_D$ to half (12 μM) or twice (48 μM) the value obtained from the fit yielded higher chi-square residuals and less good agreement between observed and simulated line shapes (*Supplementary file 3*). We also performed an independent conventional fraction bound analysis, which yielded lower and upper bounds for the $K_D$ to be 25 μM and 55 μM, respectively (*Figure 2d*, inset). Together, these data suggest that amantadine likely binds to ABP with a $K_D$ in the low micromolar range. We were not able to assign the resonances of the protein due to difficulties in preparing labeled samples, but we observe significantly affected methyl resonances which could correspond to the Ile-64, Leu-67, and Ala-68 residues in close proximity to the intended amantadine binding site in the designed structure (*Figure 1b*). Together, the NMR titrations of both amide and methyl groups suggest the presence of localized backbone and side-chain conformational changes in ABP upon amantadine binding.

## X-ray crystal structure is in close agreement with the design model

We carried out crystallographic studies to characterize the interaction between ABP and amantadine. Crystallization screen trays were set up with the same protein sample with or without ~five fold molar excess amantadine (7.5 mM). Crystals were obtained in the presence but not the absence of amantadine, consistent with ordering upon amantadine binding. The X-ray crystal structure of ABP +amantadine was solved to 1.04 Å, providing a high-resolution view of the ABP-amantadine complex structure (*Figure 3a* and *Supplementary file 2A*). The crystal structure overlays well with the design model, with an RMSD of 0.63 Å (TMAlign [*Zhang and Skolnick, 2005*]) (*Figure 3a*). The primary difference between the design model and crystal structure is in the compactness of helices in the amantadine-binding region and rotation of the amantadine molecule accompanied by the presence of crystallographic water molecules (*Figure 3a*). In the original design model, the amino group of amantadine was oriented to hydrogen bond directly to Ser-71 in ABP (*Figure 3b*), but in the X-ray crystal structure, amantadine was found to be rotated 60° with ordered water molecules mediating hydrogen bonds to Ser-71 residues in ABP (*Figure 3b–c*). Amantadine is often bound as a tri- or tetra-hydrate with the waters associated with the amino group (*Thomaston et al., 2018*; *Wang et al., 2011*), for example in a recent crystal structure of amantadine bound to the influenza M2 channel protein, amantadine bound the key His-37 residues in M2 through water-mediated interactions (*Figure 3d*). Including explicit water molecules in the Rosetta design calculations could enable the design of proteins that bind amantadine with higher affinity.

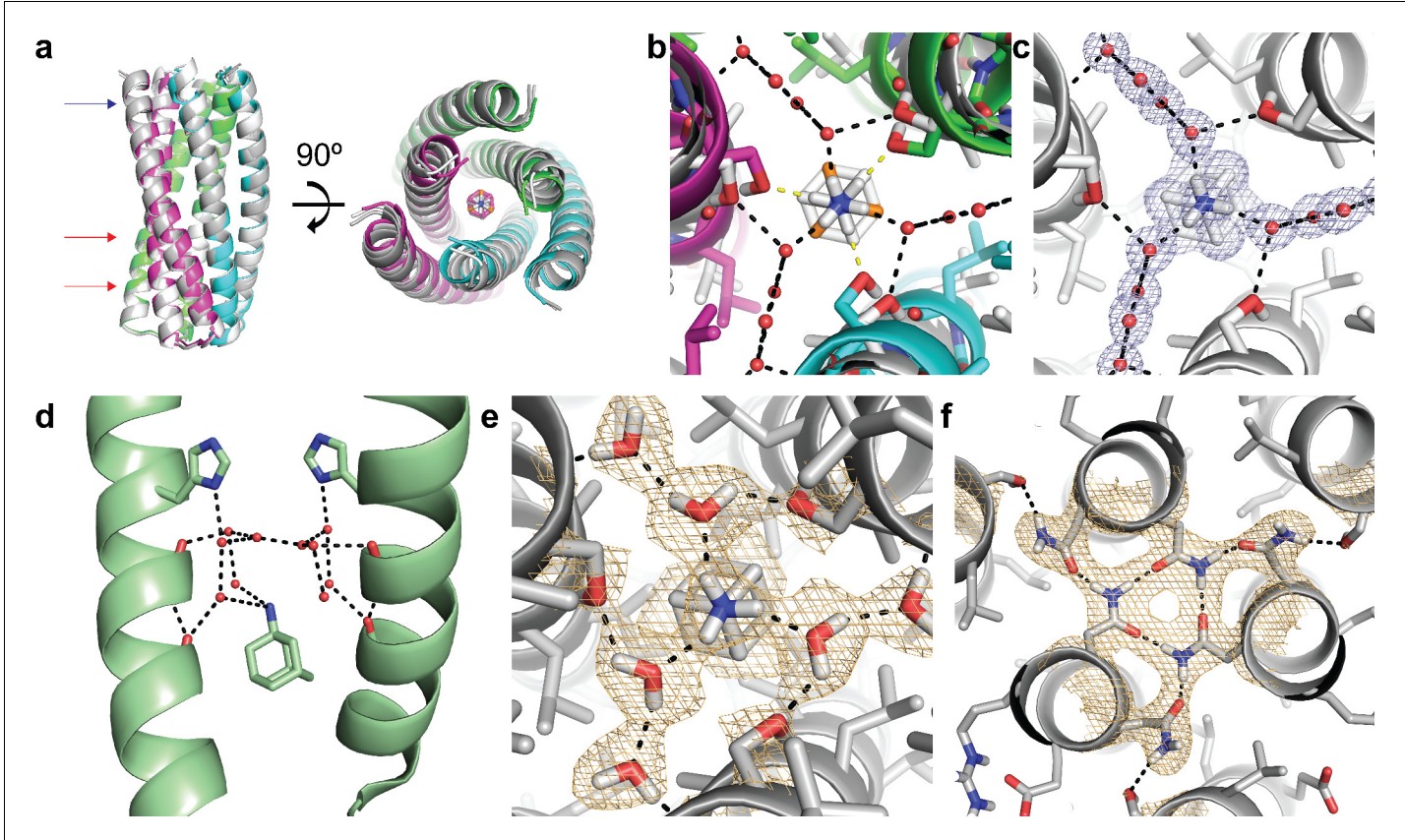

**Figure 3.** Structural characterization of the ABP-amantadine interaction. (**a**) The high-resolution X-ray structure (white) and neutron structure (gray) of ABP in complex with amantadine are very close to the computational model (magenta, green, and cyan) (RMSD of 0.63 Å and 0.59 Å, respectively). The blue arrow indicates the amantadine binding site shown in (**c,e**), and the red arrows indicate the hydrogen bond networks, one of which is shown in (**f**). (**b**) A zoomed in overlay of the X-ray structure (white) and the design model (colored) reveal a shift in the helices within the amantadine-binding region, accompanied by a ~ 60° rotation of amantadine and the presence of water molecules that that mediate hydrogen bonding to Ser-71. Yellow dashed lines show direct hydrogen bonds to Ser-71 in the design model and black dashed lines show the hydrogen bonds observed in the X-ray structure. (**c**) Clear electron density can be observed for amantadine and ordered water molecules in the binding site of ABP ($2F_o$ - $F_c$ map contoured at 1.0σ). Water-mediated hydrogen bonds are observed between Ser-71 and the amino group of amantadine (black dashed lines). (**d**) The crystal structure (pale green) of amantadine bound to the influenza M2 protein through water-mediated hydrogen bonds (image generated from PDB: 6BKK [*Thomaston et al., 2018*]). (**e**) The nuclear scattering length density map shows the positions of deuterium atoms, including two ordered water molecules that mediate the hydrogen-bond network between Ser-71 and amantadine ($2F_o$ - $F_c$ contoured at 1.0σ). Hydrogen bonds are shown as black dashed lines. (**f**) Clear nuclear scattering length density can be observed for residues involved in the designed hydrogen-bond networks (black dashed lines) in ABP ($2F_o$ - $F_c$ map contoured at 1.0σ).

The online version of this article includes the following figure supplement(s) for figure 3:

**Figure supplement 1.** Stereo images of the electron and neutron length scattering density maps for a hydrogen bond network in ABP.

## Neutron structure reveals hydrogen-bond interactions

We used neutron diffraction to directly visualize the intra- and intermolecular hydrogen-bond networks within ABP and with amantadine. Room temperature neutron data collected to 2.3 Å on a deuterium-exchanged ABP crystal revealed the protonation state of residues that form hydrogen bonds to the polar amino group of amantadine and the orientation of the $D_2O$ network that helps anchor amantadine at the trimeric interface (*Figure 3e–f*; *Supplementary file 2B*). Nuclear scattering length density is clearly visible for the deuterium atoms of water molecules DOD-31 and DOD-41, and the amino group of amantadine. DOD-31 accepts hydrogen bonds from the amino hydrogens of amantadine and donates hydrogen bonds to SER-71 and DOD-41. In both the X-ray and neutron structures, amantadine sits at a special position on the $C_3$ symmetry axis, with its adamantane moiety interlocked at the trimeric interface and surrounded by the hydrophobic residues Ile-64,

Leu-67, and Ala-68. The neutron structure shows no evidence of H/D exchange within the trimeric interface, even though the crystals were $D_2O$ exchanged, suggesting that the trimeric core is tightly packed, and that once formed, remains stable and inaccessible to solvent at room temperature.

## ABP variant designs

ABP binds amantadine in a manner very similar to the design model and leads to significant localized changes as visualized by solution NMR, but it is a constitutive (albeit perhaps not a very thermal-stable) trimer in the absence of amantadine. We sought to generate ABP derivatives with amantadine-inducible trimerization by destabilizing the trimer in a variety of ways (*Figure 1—figure supplement 1*). Sidechain truncations such as Ala/Ser mutations were introduced in the core of ABP to destabilize the trimeric interface (*Figure 1—figure supplement 1b*), but these constructs were either poorly expressed or lost amantadine binding activity as assessed by thermofluor. The Rosetta HBNet protocol that was used to generate the hydrogen-bond networks in ABP was extended to search for intermolecular hydrogen bond interactions between residues that span the monomer-monomer interfaces reduce the extent of hydrophobic packing of the trimer (*Figure 1—figure supplement 1c*), but these constructs were again poorly expressed. Truncations of the outer helix (*Figure 1—figure supplement 1d–f*) were attempted, but this resulted in mostly insoluble protein or monomeric species that no longer bound amantadine (In one case, truncation of both helices (*Figure 1—figure supplement 1g*) resulted in a tetramer that no longer bound amantadine). In larger-scale redesigns, the core amantadine-binding site was backed up with helical repeat fusions to stabilize the helical core (*Figure 1—figure supplement 1h*), but these constructs remained constitutive trimers. Taken together, these results suggest that conversion of our design into well behaved monomers that assemble into a trimer in the presence of amantadine is difficult because the subunit-subunit interface in the trimer involves considerable non-polar surface area which makes the subunits poorly behaved as monomers, and the low binding affinity for amantadine does not provide a strong driving force for assembly.

## Discussion

We report the characterization of a de novo designed trimeric protein, ABP, which binds the small molecule drug amantadine. The designed protein contains hydrogen-bond networks that specify the trimeric state and water-mediated binding to amantadine. The solution NMR data suggest that ABP adopts a stable, symmetric structure and readily binds amantadine. The high-resolution X-ray crystal structure of the designed protein in complex with amantadine is very close to the computational model, and the neutron structure demonstrates the presence of the designed hydrogen-bond networks. While we were unable to design an inducible trimer, our results are an advance for protein design as to our knowledge this is the first successful de novo design of a homo-trimeric protein that binds a $C_3$ symmetric small molecule other than a metal ion (*Mills et al., 2016*). Our results suggest two major bottlenecks to the goal of an amantadine-inducible trimerization system based on amantadine binding at a helical bundle three-fold interface: (1) Amantadine, given its small size, does not provide strong driving force for trimerization. (2) Well behaved monomers in the absence of amantadine are hard to achieve in a system with substantial buried nonpolar surface area at the trimer interface (which becomes exposed in the monomers). Success in designing protein homo-trimerization systems will likely require smaller subunit interfaces and higher affinity binding sites, perhaps using larger $C_3$ molecules.

## Materials and methods

**Key resources table**

| Reagent type (species) or resource | Designation | Source or reference | Identifiers | Additional information |
|---|---|---|---|---|
| Strain, strain background (include species and sex here) | One Shot BL21 Star (DE3) Chemically Competent *E. coli* | Invitrogen (Thermo Fisher Scientific) | C601003 | |

*Continued on next page*

*Continued*

| Reagent type (species) or resource | Designation | Source or reference | Identifiers | Additional information |
|---|---|---|---|---|
| Recombinant DNA reagent | pET28b(+) DNA - Novagen | Sigma-Aldrich (Millipore Sigma) | 69865–3 | |
| Commercial assay or kit | NeXtal Tubes JCSG+Suite | Qiagen | 130720 | |
| Chemical compound, drug | Amantadine hydrochloride | Sigma-Aldrich (Millipore Sigma) | A1260 | |
| Software, algorithm | Rosetta software suite | Rosetta Commons | N/A | |

## Rosetta design

Design calculations were performed using RosettaDesign. The Rosetta software suite is available free of charge to academic users and can be downloaded from http://www.rosettacommons.org. Instructions and inputs for running these applications, and all other data necessary to support the results and conclusions (including the.xml,. cst,. params, and in.res files mentioned below), are provided in *Supplementary files 1A-1D*.

The initial 2LC3H6_13 scaffold was previously generated using parametric design (*Boyken et al., 2016*). Briefly, the parametrically generated backbone was regularized using cartesian space minimization in Rosetta and a special instance of the HBNet protocol - HBNetStapleInterface - was used to identify combinations of hydrogen-bond networks. The helices of monomer subunits were connected into a single chain and the assembled proteins were designed using symmetric Rosetta sequence design calculations in $C_3$ symmetry.

In order to create the amantadine binding site, the RosettaScripts protocol was used with user-defined design of the residue positions within 15 Å of the ligand (.xml). A Rosetta constraint (.cst) file was used to specify the atom-pair constraints in amantadine. A molecule parameter (.params) file was generated for amantadine in RosettaDesign. Amantadine was split into one third, and the nitrogen and carbon atoms on the axis of rotation were virtualized. Rotamers were repacked with Layer-Design and resfile types (in.res) were used to specify Ser/Thr at residue positions hydrogen-bonding to amantadine.

## Cloning, protein expression and purification

ABP was cloned into the pET28b(+) vector at NdeI and XhoI restriction sites. The resulting expressed protein sequence was as follows:

MGSSHHHHHHSSGLVPR/GSHMG//DAQDKLKYLVKQLERALRELKKSLDELERSLEELEKNPSEDALVE NNRLNVENNKIIVEVLRIILELAKASAKLA

where '/' demarks a thrombin cleavage site and '//' demarks the beginning of the designed sequence within Rosetta and the numbering of amino acids within this manuscript.

Constructs were transformed into BL21-Star (DE3) competent cells (Life Technologies). Cells harboring the plasmid were grown at 37°C in Terrific Broth II medium containing a final concentration of 0.05 mg/ml kanamycin. Once cells reached an OD600 of 0.6–0.8, cells were cooled to 18°C and induced with 0.25 mM IPTG overnight. After this period, cells were harvested by centrifugation at 4000 r.p.m. for 10 min at 4°C. Cell pellets were resuspended in 60 ml of 25 mM Tris (pH 8.0), 300 mM NaCl, 20 mM imidazole (pH 8.0), and 1 mM PMSF per 1 L of Terrific Broth II medium and stored at −80°C.

Cells were thawed in the presence of 0.25 mg/ml lysozyme and disrupted using sonication on ice for 60 s. The cell extract was obtained by centrifugation at 13,000 r.p.m. for 30 min at 4°C and was applied onto Ni-NTA agarose beads (Qiagen) equilibrated with wash buffer (25 mM Tris (pH 8.0), 300 mM NaCl, and 20 mM imidazole (pH 8.0)). The wash buffer was used to wash the nickel column three times with five column volumes. After washing, protein was eluted with five column volumes of elution buffer (wash buffer with 300 mM imidazole).

The eluate was buffer-exchanged with SAXS buffer (25 mM Tris (pH 8.0), 150 mM NaCl, and 2% glycerol) to lower the imidazole concentration from ~300 mM to <20 mM and cleaved with

restriction-grade thrombin (EMD Millipore 69671–3) overnight at 20°C. After overnight cleavage, the sample was flowed over equilibrated Ni-NTA agarose beads and the flow-through was captured.

The protein sample was further purified by gel chromatography using a Superdex 75 Increase 10/300 GL column (GE Healthcare) equilibrated with SAXS buffer. The fractions containing the protein of interest were pooled and concentrated using a 3 K MWCO Amicon centrifugal filter (Millipore).

## Thermofluor assay

Thermofluor assays were performed in SAXS buffer using a CFX96 Touch Real-Time PCR machine (Bio-Rad). Thermal stability assays were performed using 45 µl of 5 µM protein (with or without 1 mM amantadine) and 5 µL of freshly prepared 200X SYPRO orange (Thermo-Fisher) solution in SAXS buffer. The temperature was ramped from 25°C to 95°C in 0.5°C increments with intervals of 5 s. Fluorescence was read in the FRET scanning mode. The average of three replicates of buffer + SYPRO orange solution (no protein control) was subtracted from the average of three replicates for each sample.

## Circular Dichroism

CD wavelength scans (260 to 195 nm) and temperature melts (25°C to 95°C) were measured using a JASCO J-1500 or an AVIV model 420 CD spectrometer. Temperature melts monitored absorption signal at 222 nm and were carried out at a heating rate of 4 °C/min. Protein samples were prepared at 0.25 mg/mL in phosphate buffered saline (PBS) pH 7.4 in a 0.1 cm cuvette.

## Solution NMR

Isotopically labeled ABP (U-[$^{15}$N] Ala $^{13}$Cβ, Ile $^{13}$Cδ1, Leu $^{13}$Cδ1/$^{13}$Cδ2, Val $^{13}$Cγ1/$^{13}$Cγ2) methyl) was prepared using well-established protocols (*Tzeng et al., 2012*) and buffer exchanged into NMR buffer (50 mM NaCl, 20 mM sodium phosphate pH 6.5, 10% (v/v) D$_2$O). Two-dimensional $^1$H-$^{15}$N SOFAST-HMQC and $^1$H-$^{13}$C SOFAST-HMQC spectra of 250 µM ABP were recorded without amantadine and with 2 mM amantadine at a $^1$H field of 800 MHz at 37°C. The pH was monitored to ensure that there were no pH changes that influence NMR shifts upon addition of amantadine hydrochloride. NMR titrations were performed using 118 µM ABP with 2D $^1$H-$^{13}$C SOFAST-HMQC experimental readouts at a $^1$H field of 800 MHz at 37°C with ABP:amantadine molar ratios of 1:0, 1:0.58, 1:1.35 and 1:2.47. Titration experiments were recorded with 16 scans with 38 msec acquisition in the indirect $^{13}$C dimension and an interscan delay of 0.2 s. Data were processed with a 4 Hz and 10 Hz Lorentzian line broadening in the direct and indirect dimensions, respectively, and fit using a two-state binding model in TITAN (*Waudby et al., 2016*) with bootstrap error analysis of 100 replicas. Identification of methyl group types (ALA, ILE, LEU) was possible due to the unique chemical shift positions of these methyl group types as referenced in the Biological Magnetic Resonance Data Bank (http://www.bmrb.wisc.edu/). All NMR data were processed with NMRPipe (*Delaglio et al., 1995*) and analyzed using NMRFAM-SPARKY (*Lee et al., 2015*).

## Crystallization of ABP

Purified ABP sample was concentrated to approximately 13 mg/ml in SAXS buffer and incubated with 7.5 mM amantadine (~five fold molar excess). Samples were screened using the sparse matrix method (*Jancarik and Kim, 1991*) with a Phoenix Robot (Art Robbins Instruments, Sunnyvale, CA) utilizing the following crystallization screens: Morpheus (Molecular Dimensions), JCSG+ (Qiagen), and Index (Hampton Research). Crystals were obtained in crystallization condition JCSG+ B9: 0.1 M Citric Acid (4.0), 20% w/v PEG 6000 (final pH 5.0). Crystals were obtained after 1 to 14 days by the sitting-drop vapor-diffusion method with the drops consisting of a 1:1 mixture of 0.2 µL protein solution and 0.2 µL reservoir solution.

## X-ray diffraction collection and structure determination of ABP

ABP crystals were placed in a reservoir solution containing 20% (*v/v*) glycerol, and then flash-cooled in liquid nitrogen. The X-ray data sets were collected at a wavelength of 1 Å at the Beamline 19-ID of the Advanced Photon Source (APS) at Argonne National Laboratory (ANL). Data sets were indexed and scaled using HKL2000 (*Otwinowski and Minor, 1997*). All the design structures were determined by the molecular-replacement method with the program *PHASER* (*McCoy et al., 2007*)

within the *Phenix* suite (*Adams et al., 2010*) using the design models as the initial search model. The atomic positions obtained from molecular replacement and the resulting electron density maps were used to build the design structures and initiate crystallographic refinement and model rebuilding. Structure refinement was performed using the *phenix.refine* (*Afonine et al., 2010*) program. Manual rebuilding using *COOT* (*Emsley and Cowtan, 2004*) and the addition of water molecules allowed construction of the final models. Root-mean-square deviation differences from ideal geometries for bond lengths, angles and dihedrals were calculated with *Phenix* (*Adams et al., 2010*). The overall stereochemical quality of all final models was assessed using the program *MOLPROBITY* (*Davis et al., 2007*). The model showed 100% of the residues in favorable regions of the Ramachandran plot with 0% outliers. Figures were prepared with *Pymol* (Pymol Molecular graphics System, Version 2.0; Schrodinger, LLC). Summaries of diffraction data and refinement statistics are provided in *Supplementary file 2A and a* stereo image of a representative region of the electron density map is shown in *Figure 3—figure supplement 1a*.

## Neutron diffraction collection and structure refinement

Neutron sized ABP crystals were obtained by seeding/feeding techniques. Crystals were grown at 20°C in sitting drops containing 3 µl protein solution mixed with 2 µl of the precipitant well solution, which contained 9% PEG 6000, 0.1 M Citric acid pH 4.0. After 20 days, a 300-micron crystal was transferred to a fresh drop containing 5 µl of precipitant and 3 µl of protein solution. After growth terminated (20 days), 2–3 µl of protein solution was fed to the drop every two weeks. When the crystal volume reached 0.2 mm$^3$ (5 months), the well solution was replaced with a $D_2O$ containing mother liquor 5 times to exchange labile hydrogen atoms with deuterium (4 months). The crystal was mounted in a quartz capillary for data collection. Neutron diffraction data were recorded using the IMAGINE instrument (*Meilleur et al., 2013*) at the High Flux Isotope Reactor at Oak Ridge National Laboratory (ORNL). A total of 17 images at 34 hr exposure were collected from two crystal orientations with 10° step intervals using a broad bandpass (2.8–4.5 Å) quasi-Laue beam. Laue images were indexed and integrated using the *LAUEGEN* (*Campbell et al., 1998*) suite of programs from CCP4; wavelength normalized using *LSCALE* (*Arzt et al., 1999*) to account for the spectral distribution of the quasi-Laue beam and then scaled and merged using *SCALA* (*Winn et al., 2011*).

A room temperature X-ray diffraction data set was collected on a smaller crystal grown under the same conditions and mounted in a quartz capillary at 293 K on a Rigaku micromax-007 HF X-ray generator with a Raxis IV++ image plate detector. The X-ray crystal structure was solved to a resolution of 1.9 Å using *Phenix* and manual model building was performed using *COOT*. Isomorphous replacement for neutron dataset was performed using Phenix with the protein model obtained from the 1.9 Å X-ray crystal structure followed by several cycles of atomic position and occupancy refinement. The overall stereochemical quality of all final models was assessed using the program *MOLPROBITY* (*Davis et al., 2007*). The model showed 98.6% of the residues in favorable regions of the Ramachandran plot and 1.4% in the allowed region. Figures were prepared with *Pymol* (Pymol Molecular graphics System, Version 2.0; Schrodinger, LLC). Summaries of diffraction data and refinement statistics are listed in *Supplementary file 2B and A* stereo image of a representative region of the neutron length scattering density map is shown in *Figure 3—figure supplement 1b*.

## Data availability

The atomic coordinates and structure factors for the X-ray and neutron crystal structures of ABP have been deposited in the RCSB Protein Data Bank under accession codes: 6N9H and 6NAF respectively. All other data generated or analyzed during this study are included in this published article (and its Supplementary files) or are available from the corresponding author on reasonable request.

## Acknowledgements

We thank Dave Roberts (DePauw University) and Norma Dukes (SBC) for assistance with X-ray diffraction data collection and data processing. Results shown in this report are derived from work performed at Argonne National Laboratory, Structural Biology Center (SBC) at the Advanced Photon Source. SBC-CAT is operated by UChicago Argonne, LLC, for the US Department of Energy, Office of Biological and Environmental Research under contract DE-AC02-06CH11357. Neutron diffraction

data were collected at the High Flux Isotope Reactor, a DOE Office of Science User Facility operated by the Oak Ridge National Laboratory. NMR data acquisition was supported through the Office of the Director, NIH, under High End Instrumentation (HIE) Grant S10OD018455, which funded the 800 MHz NMR spectrometer at UCSC. JP is supported by the Washington Research Foundation Innovation Postdoctoral Fellowship. ACM and N.GS are supported by an R35 Outstanding Investigator Award through NIGMS(1R35GM125034-01). SEB was supported by the Burroughs Wellcome Fund Career Award at the Scientific Interface. This work was also supported by HHMI. The content is solely the responsibility of the authors and does not necessarily represent the official views of the funding agencies.

## Additional information

### Competing interests

Jooyoung Park, Scott E Boyken, Kathy Y Wei, Gustav Oberdorfer, David Baker: JP, SEB, KYW, GO, and DB have filed a provisional application based for "Amantadine Binding Protein" (Application # 62/834,592). The other authors declare that no competing interests exist.

### Funding

| Funder | Grant reference number | Author |
| --- | --- | --- |
| Washington Research Foundation | Innovation Postdoctoral Fellowship | Jooyoung Park |
| National Institute of General Medical Sciences | 1R35GM125034-01 | Andrew C McShan Nikolaos G Sgourakis |
| Burroughs Wellcome Fund | Career Award at the Scientific Interface | Scott E Boyken |
| Howard Hughes Medical Institute | | David Baker |

The funders had no role in study design, data collection and interpretation, or the decision to submit the work for publication.

### Author contributions

Jooyoung Park, Conceptualization, Methodology, Software,Validation, Formal Analysis, Investigation, Resources, Writing – Original Draft Preparation, Writing – Review & Editing, Visualization, Project Administration, Funding Acquisition; Brinda Selvaraj, Conceptualization, Methodology, Validation,Formal Analysis, Investigation, Writing – Original Draft Preparation, Writing –Review & Editing, Visualization; Andrew C McShan, Conceptualization, Methodology, Validation, Formal Analysis,Investigation, Writing – Original Draft Preparation, Writing – Review &Editing, Visualization, Funding Acquisition; Scott E Boyken, Conceptualization, Methodology, Software, Validation,Resources, Writing – Review & Editing, Funding Acquisition; Kathy Y Wei, Gustav Oberdorfer, Conceptualization, Methodology, Software, Validation, Writing – Review & Editing; William DeGrado, Conceptualization, Validation, Writing – Review & Editing; Nikolaos G Sgourakis, Conceptualization, Software, Validation, Formal Analysis, Writing – Original Draft Preparation, Writing – Review & Editing, Project Administration, Funding Acquisition; Matthew J Cuneo, Dean AA Myles, Conceptualization, Validation, Writing – Review & Editing, Project Administration; David Baker, Conceptualization, Validation, Writing – Original Draft Preparation, Writing – Review & Editing, Visualization, Supervision, Project Administration, Funding Acquisition

### Author ORCIDs

Jooyoung Park (iD) https://orcid.org/0000-0001-8557-641X
Andrew C McShan (iD) https://orcid.org/0000-0002-3212-9867
Scott E Boyken (iD) https://orcid.org/0000-0002-5378-0632
Kathy Y Wei (iD) https://orcid.org/0000-0002-8794-1385
Nikolaos G Sgourakis (iD) https://orcid.org/0000-0003-3655-3902

Matthew J Cuneo ⬛ https://orcid.org/0000-0002-1475-6656
Dean AA Myles ⬛ https://orcid.org/0000-0002-7693-4964
David Baker ⬛ https://orcid.org/0000-0001-7896-6217

Decision letter and Author response
Decision letter https://doi.org/10.7554/eLife.47839.sa1
Author response https://doi.org/10.7554/eLife.47839.sa2

## Additional files

### Supplementary files

• Supplementary file 1. *Supplementary file 1A* RosettaScripts XML file (.xml). Sample RosettaScripts XML file *Supplementary file 1B* Parameter constraint file for amantadine (.cst). Parameter constraint file for amantadine used in the RosettaDesign calculations. *Supplementary file 1C* Parameter definition file for amantadine (.params). Parameter definition file for amantadine used in the RosettaDesign calculations. *Supplementary file 1D* Restype file (in.res). Restype file used in the RosettaDesign calculations.

• Supplementary file 2. *Supplementary file 2A* X-ray data collection and refinement statistics. Data collection and refinement statistics for the X-ray structure of ABP *Supplementary file 2B* Neutron scattering data collection and refinement statistics Data collection and refinement statistics for the neutron and room temperature X-ray structure of ABP.

• Supplementary file 3. NMR line shape fitting analysis with fixed $K_D$ values.

• Transparent reporting form

### Data availability

The atomic coordinates and structure factors for the X-ray and neutron crystal structures of ABP have been deposited in the RCSB Protein Data Bank under accession codes: 6N9H and 6NAF respectively. All other data generated or analyzed during this study are included in this published article (and its Supplementary files).

The following datasets were generated:

| Author(s) | Year | Dataset title | Dataset URL | Database and Identifier |
|---|---|---|---|---|
| Park J, Baker D | 2018 | De novo designed homo-trimeric amantadine-binding protein | http://www.rcsb.org/structure/6N9H | RCSB PDB, 6N9H |
| Selvaraj B, Park J, Cuneo MJ, Myles DAA, Baker D | 2018 | De novo designed homo-trimeric amantadine-binding protein | http://www.rcsb.org/structure/6NAF | RCSB PDB, 6NAF |

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
