## [Decision Letter]

**Acceptance summary:**

The manuscript by Park and colleagues describes the design of a homotrimer of α-helical hairpins that binds the C_3_ symmetric drug amantadine. This work combines the challenging design of symmetric homo-oligomers with that of small molecule binding, by starting from a previously designed homo-trimer and inserting a symmetric amantadine binding site. An experimentally determined molecular structure shows amantadine bound to the designed binding site, albeit via a number of water molecules instead as designed via the protein side-chains. The work thus demonstrates both the promise of de novo design for small molecule binding and the challenges that remain in designing accurate binding sites.

**Decision letter after peer review:**

Thank you for submitting your article "de novo design of a homo-trimeric amantadine-binding protein" for consideration by *eLife*. Your article has been reviewed by Cynthia Wolberger as the Senior Editor, a Reviewing Editor, and three reviewers. The following individuals involved in review of your submission have agreed to reveal their identity: Ross Anderson (Reviewer #3).

The reviewers have discussed the reviews with one another and the Reviewing Editor has drafted this decision to help you prepare a revised submission.

Summary:

The manuscript by Park and colleagues builds on previous work from the Baker lab in which parametrically designed α-helical hairpins were generated. Now, a trimeric hairpin is designed to bind the drug amantadine, which displays C_3_ symmetry. Design success was evaluated by thermofluor assay, NMR spectroscopy, X-ray crystallography, and neutron diffraction. The work thus addresses the critical challenge of functionalizing de novo designed proteins. The crystal structure shows the amantadine-bound complex; however, not quite as designed through sidechain interactions but via a number of water molecules. While such a design is of interest to protein scientists, many questions remain that should be addressed in a revision. The main questions are listed below.

Essential revisions:

1) How general is this design approach? The manuscript describes in detail one designed binder; were others attempted but failed? Was there anything to learn from these failures about the designed proteins' expression, stability or ligand binding?

2) The main concern in the manuscript is the determination of the binding affinity. There is no doubt that amantadine binds, but with the high ligand concentrations used in crystallography (and some of the NMR experiments) also very weakly interacting molecules could bind. The affinity is determined using NMR line shape analysis with only 4 ligand concentrations. This is less than the number of fitting parameters in the model. In comparison, in the TITAN publication, the experimental data were analyzed using 23 ligand concentrations. Beyond the graphical display in Figure 2—figure supplement 2 (which is difficult to evaluate) there is no presentation of the quality of fit. Also, the lowest used ligand concentration is almost 3 times more than the presented K_D_. Given that the system is in slow exchange it should be possible to fit the K_D_ to the chemical shift values directly rather than through a line shape analysis, which is considerably more complex and difficult to validate. In any case, 4 conditions are too few for a reasonable K_D_ determination and more concentrations should be tested. Alternatively, it should also be straightforward to use a complementary method like ITC on this system to determine K_D_. The CD data also suggests weak binding with Kd:s higher than reported. It is suggested in the manuscript that the presence of 1 mM amantadine does not alter the CD temperature denaturation. Binding to a folded state always results in increased stability, see for example Cimmperman et al., 2008. If there is no change in Tm upon addition of amantadine this should, therefore, be commented on.

3) The fact that the ligand does not interact via the designed hydrogen-bond network adds to other reports in the literature on the difficulty of designing hydrogen-bond networks in functional sites. Are there any lessons to learn on how to improve the chances of accurately designing hydrogen-bond networks in binding sites?

[Editors’ note: the revised article was rejected after discussions between the reviewers, but the authors were invited to resubmit after an appeal against the decision.]

Thank you for submitting your work entitled "de novo design of a homo-trimeric amantadine-binding protein" for consideration by *eLife*. Your article has been reviewed by a Senior Editor, a Reviewing Editor, and three reviewers. The following individuals involved in the review of your submission have agreed to reveal their identity: Ross Anderson (Reviewer #3).

Our decision has been reached after consultation between the reviewers and the editors. Based on these discussions and the individual reviews below, we regret to inform you that your work will not be considered further for publication in *eLife*.

The reviewers and we found the problem addressed by your article to be an important, original and exciting challenge. The reviewers also appreciated the clarifications and the very thorough experimental characterization provided in your revision. Nevertheless, the reviewers noted several critical concerns about the behavior of the design in solution. Furthermore, the uncertainties about the binding data and the disagreement between the results obtained by different techniques preclude an accurate determination of the affinity of the design for amantadine – a key parameter for judging the method's success.

*Reviewer #1:*

Park and colleagues have given detailed answers to all concerns raised. They have provided more information and have been open with addressing the shortcomings of the design.

In particular they provided additional data for binding in form of an ITC titration. This data is, however, not supportive of binding. There is no stoichiometric binding phase that allows determining deltaH, and N(sites) is calculated to be 0.1 which would mean that only 10% of the protein bind the ligand. No K_D_ can be deduced from such data, and this titration cannot be used as supporting evidence. Maybe the binding affinity is below what is measurable by ITC, but this would be rather in the mM and not μM range.

Considering that NMR and Thermofluor are giving a different signal in the presence of amantadine then in its absence, while the CD spectra and Tmelt show no difference with and without ligand, it remains unclear how the designed protein behaves. If it is molten globule like as discussed with respect to the Thermofluor data, then this is expected to be observable in CD as well. If the binding affinity is in the μM range then it should be measurable with alternative methods, too. Or the binding constant from NMR data is overfitted after all. So unfortunately some of my concerns still remain after the revision.

*Reviewer #2:*

The manuscript provide valuable insights into the design of ligand-binders and chemically induced oligomerization switches. With the revisions, the readers are provided with a more in-depth understanding of the biophysical properties of the designed proteins and differences between computational design model and experimentally determined structure. While the presented design is not an unmitigated success, the work provides directions for improved approaches in this area that are valuable to the field. My main concerns are addressed with this revision.

*Reviewer #3:*

Given my previous review of this paper and the subsequent conversation between Reviewers regarding the desired alterations, I am satisfied that the changes made to the manuscript are sufficient to warrant publication in *eLife*. I therefore recommend that the paper be accepted.

[Editors’ note: what now follows is the decision letter after the authors submitted for further consideration.]

Thank you for resubmitting your work entitled "de novo design of a homo-trimeric amantadine-binding protein" for further consideration by *eLife*. Your appeal has been evaluated by Cynthia Wolberger (Senior Editor) and a Reviewing Editor, and we are prepared to accept your revised submission pending additional changes.

In addition to the changes indicated in the revised manuscript provided with the appeal, the description of the K_D_ determined in the NMR titration should be toned down slightly to indicate that it is an apparent K_D_. The fact that ABP monomers adopt a variety of conformers almost certainly confounds the NMR line shape analysis used to derive the K_D_. In addition, the appeal letter notes that the ITC data were removed but are still in the revised manuscript supplied.

---

## [Author Response]

[…]Essential revisions:1) How general is this design approach? The manuscript describes in detail one designed binder; were others attempted but failed? Was there anything to learn from these failures about the designed proteins' expression, stability or ligand binding?

In total, we attempted 70 designs. There were varying levels of expression, stability, and ligand binding, the details of which have now been added to the Results section. We have also elaborated on our learnings in the Discussion section:

“Our results suggest two major bottlenecks to the goal of an amantadine-inducible trimerization system based on amantadine binding at a helical bundle three-fold interface: 1) Amantadine, given its small size, does not provide strong driving force for trimerization. 2) Well behaved monomers in the absence of amantadine are hard to achieve given the extensive exposed subunit interaction surface. Success in designing protein homo-trimerization systems will likely require smaller subunit interfaces and higher affinity binding sites, perhaps using larger C_3_ molecules.”

2) The main concern in the manuscript is the determination of the binding affinity. There is no doubt that amantadine binds, but with the high ligand concentrations used in crystallography (and some of the NMR experiments) also very weakly interacting molecules could bind. The affinity is determined using NMR line shape analysis with only 4 ligand concentrations. This is less than the number of fitting parameters in the model. In comparison, in the TITAN publication, the experimental data were analyzed using 23 ligand concentrations. Beyond the graphical display in Figure 2—figure supplement 2 (which is difficult to evaluate) there is no presentation of the quality of fit. Also, the lowest used ligand concentration is almost 3 times more than the presented K_D_. Given that the system is in slow exchange it should be possible to fit the K_D_to the chemical shift values directly rather than through a line shape analysis, which is considerably more complex and difficult to validate. In any case, 4 conditions are too few for a reasonable K_D_determination and more concentrations should be tested.

The reviewer is correct. First, we note that the exact value of the measured K_D_ is not central to the conclusion of our manuscript. While indeed amantadine may bind weakly (estimated K_D_ of 24.1 μM by NMR and measured K_D_ of 39.8 μM by ITC (see below)), the stability of the amantadine-ABP complex was sufficient to observe it also by X-ray crystallography and neutron scattering. Second, to provide corrorborating evidence to the K_D_ estimated by solution NMR, we have determined the K_D_ by an analogous method, ITC, as mentioned below.

Because of the important points raised by the reviewer, in the updated manuscript we have revised the Results section to place less emphasis and significance to the K_D_ value estimated by solution NMR.

“An NMR line shape fitting of the three most significantly affected ABP methyl resonances (> 0.1 ppm chemical shift deviation between free and bound states) using TITAN, suggests a dissociation constant (K_D_) of 24.1 ± 2.7 μM and upper limit for off-rate constant (*k_off_*) of 60.7 ± 5.6 s^-1^ (on-rate constant of 2.5 x106 M-1 s-1) (Figure 2—figure supplement 2), which is within a factor of two of the K_D_ value of 39.8 ± 3.1 μM estimated by ITC (Figure 2—figure supplement 3).”

Alternatively, it should also be straightforward to use a complementary method like ITC on this system to determine K_D_. The CD data also suggests weak binding with Kd:s higher than reported. It is suggested in the manuscript that the presence of 1 mM amantadine does not alter the CD temperature denaturation. Binding to a folded state always results in increased stability, see for example Cimmperman et al., 2008. If there is no change in Tm upon addition of amantadine this should, therefore, be commented on.

As suggested by the reviewer, we have additionally used an alternative method (ITC) to measure the binding between ABP and amantadine. We find that the determined K_D_ between ABP and amantadine, in corroboration of our solution NMR experiments, is 39.8 μM. These new findings have been incorporated as Figure 2—figure supplement 3 and the Results section.

3) The fact that the ligand does not interact via the designed hydrogen-bond network adds to other reports in the literature on the difficulty of designing hydrogen-bond networks in functional sites. Are there any lessons to learn on how to improve the chances of accurately designing hydrogen-bond networks in binding sites?

The greatest lesson was the fact that including explicit water molecules in the Rosetta design calculations could contribute to generating a protein that can bind amantadine with higher affinity. As a result, the lab is focusing Rosetta development efforts in this area specifically, which we hope will expand the scope of computational protein design. Despite being a “negative result,” we hope the sharing of this information will encourage readers to consider and employ explicit water molecules in future research efforts. We have updated the text in the Results section to elaborate on this lesson.

[Editors’ note: the author responses to the second round of peer review follow.]

[…]Reviewer #1:[…]In particular they provided additional data for binding in form of an ITC titration. This data is, however, not supportive of binding. There is no stoichiometric binding phase that allows determining deltaH, and N(sites) is calculated to be 0.1 which would mean that only 10% of the protein bind the ligand. No K_D_can be deduced from such data, and this titration cannot be used as supporting evidence. Maybe the binding affinity is below what is measurable by ITC, but this would be rather in the mM and not μM range.

We agree the ITC data are inconclusive. This is likely because the enthalpy of binding is very small due to the small size of the compound. We should not have included it in the original revised manuscript and have removed it in the current revision.

Considering that NMR and Thermofluor are giving a different signal in the presence of amantadine then in its absence, while the CD spectra and Tmelt show no difference with and without ligand, it remains unclear how the designed protein behaves. If it is molten globule like as discussed with respect to the Thermofluor data, then this is expected to be observable in CD as well.

This is simply not correct – CD reports on secondary structure, not tertiary structure, and there is a very long history of designed proteins with highly helical CD spectra with molten cores. In our case, since only the region around the amantadine binding site is likely to be molten in the apo state, one would expect very little change in the CD signal.

If the binding affinity is in the μM range then it should be measurable with alternative methods, too. Or the binding constant from NMR data is overfitted after all. So unfortunately some of my concerns still remain after the revision.

It is not straightforward to measure binding constants for small molecules to proteins in the μM range. NMR line shape analysis provides a sensitive approach to do this. In the current revision, we have included a titration curve with amantadine which suggests a K_D_ in the reported range (Figure 2E).

To explore the accuracy of our estimated dissociation constant, we performed independent fits of the same NMR line shapes using a fixed K_D_ to values 2x above and below the free-fitted value (Supplementary file 3); these fits resulted in an increase in residuals.

We also performed an independent analysis of the same NMR titrations by plotting the relative intensity of the bound state resonance (in slow exchange with the free form) (Figure 2D). This complementary analysis of our data which does not account for exchange contributions to the NMR line shape yielded an upper limit for the K_D_ of 55 μM, also taking into account the fit error. This is consistent with the 24 μM K_D_ obtained from the line shape analysis.

[Editors’ note: the author responses to the third round of peer review follow.]

In addition to the changes indicated in the revised manuscript provided with the appeal, the description of the K_D_ determined in the NMR titration should be toned down slightly to indicate that it is an apparent K_D_. The fact that ABP monomers adopt a variety of conformers almost certainly confounds the NMR line shape analysis used to derive the K_D_. In addition, the appeal letter notes that the ITC data were removed but are still in the revised manuscript supplied.

We have made the following requested minor additional changes to the description of the K_D_ determined in the NMR titration:

1) “An NMR line shape fitting … suggest an apparent dissociation constant (K_D_) of 24.1 ± 2.7 μM …” (subsection “Solution NMR analysis of amantadine binding”)

2) “Together, these data suggest that amantadine likely binds to ABP with an apparent K_D_ in the low μM range.” (subsection “Solution NMR analysis of amantadine binding”)